# The structure of a tetrameric septin complex reveals a hydrophobic element essential for NC-interface integrity

Benjamin Grupp[1], Lukas Denkhaus[2], Stefan Gerhardt [2], Matthis Vögele[1], Nils Johnsson [1] & Thomas Gronemeyer [1✉]

The septins of the yeast *Saccharomyces cerevisiae* assemble into hetero-octameric rods by alternating interactions between neighboring G-domains or N- and C-termini, respectively. These rods polymerize end to end into apolar filaments, forming a ring beneath the prospective new bud that expands during the cell cycle into an hourglass structure. The hourglass finally splits during cytokinesis into a double ring. Understanding these transitions as well as the plasticity of the higher order assemblies requires a detailed knowledge of the underlying structures. Here we present the first X-ray crystal structure of a tetrameric Shs1-Cdc12-Cdc3-Cdc10 complex at a resolution of 3.2 Å. Close inspection of the NC-interfaces of this and other septin structures reveals a conserved contact motif that is essential for NC-interface integrity of yeast and human septins in vivo and in vitro. Using the tetrameric structure in combination with AlphaFold-Multimer allowed us to propose a model of the octameric septin rod.

[1] Institute of Molecular Genetics and Cell Biology, Ulm University, Ulm, Germany. [2] Institute of Biochemistry, Albert-Ludwigs University, Freiburg, Germany.
✉email: thomas.gronemeyer@uni-ulm.de

Septins are present in all mammalian and fungal cellular systems and participate in so diverse intracellular processes such as cytokinesis, polarity establishment or cellular adhesion[1]. They are cytoskeletal guanine nucleotide-binding proteins that belong to the superclass of P-loop NTPases. Unlike other small GTPases from the Ras family, they do not act as signal transduction molecules. Instead, septins are scaffolds that recruit other proteins to sites of their activities[2–4]. By binding to phospholipids, septin higher order structures might also act as diffusion barriers that compartmentalize the plasma membrane into functionally distinct domains[5,6].

All septins share a central guanine nucleotide-binding domain (short G-domain) that is flanked by variable N- and C-terminal extensions. The G-domain contains all structural features of small GTPases like the P-loop and the distinct switch 1 and switch 2 loops including the invariant DXXG-motif (Fig. 1B, C)[7]. Septin-specific features are located N-terminally to the β1 sheet and C-terminally to the α-helix. These include a polyacidic region (PAR), following the α4-helix and leading into the short α5'-helix which connects to the β6-sheet. The following septin unique element (SUE) is a prominent feature of all septin G-domains and is folded into a characteristic β-meander consisting of three β-strands (SUE-βββ) and the C-terminal helices α5 and α6[8]. All septins possess N-terminally to the G-domain an additional α0-helix. C-terminally, they possess a coiled-coil extension which is thought to be involved in filament bundling[9], except the septins of subgroup 1a (Cdc10, SEPT9, SEPT3, SEPT12)[10] which do not have a C-terminal extension.

Septins assemble into hetero-octameric rods and further into apolar filaments through alternating interactions between two G-domain cores (G-interface) or between two G-domain N- and C-termini (NC-interface), respectively[7,11,12]. The canonical septin octamer of the baker's yeast *Saccharomyces cerevisiae* septins is formed by the linear arrangement of the subunits Cdc11-Cdc12-Cdc3-Cdc10-Cdc10-Cdc3-Cdc12-Cdc11[13]. A subset of the rods display the subunit Shs1 at their terminal positions instead of Cdc11[14]. Reconstitution experiments show that Shs1-rods form curved fibers below membranes instead of the more linear fibers of the Cdc11-containing rods[15].

Binding of GTP or GDP was documented for most and hydrolysis for some septin subunits but GTPase activity and the role of nucleotide binding are still poorly understood[16]. Current evidence favors a role of nucleotide binding in the formation of higher ordered septin structures[17] and in contributing to G-interface stability[18].

The PDB database lists currently 23 entries from various human septin monomers, hetero-dimers or homo-multimers and two entries from the SEPT2/6/7 trimer and hexamer, respectively[7,19]. This large pool of structural information enabled to predict the mode of GTP hydrolysis in some of the human septins and the architecture of the interfaces[8,20].

Whereas all structures of single human septins contain at least a homo-dimer with bound nucleotide in the asymmetric unit, the yeast Cdc11 adopts the structure of a monomeric apo protein[21]. The lack of structural information for yeast septins impedes a better mechanistic understanding of the formation of yeast septin complexes, their dynamic transitions as well as their observed structural plasticity.

We present herein the crystal structure of a hetero-tetrameric yeast septin complex and identify a so far neglected structural element within the NC-interface that is essential for septin rod assembly from yeast to man.

Using this new structural information, AlphaFold Multimer[22] enabled us to assemble the complete septin octamer from two identical tetramers.

## Results

### Crystal structure of a tetrameric Shs1 septin complex from yeast.

The octamer of the baker's yeast *Saccharomyces cerevisiae* septins consisting of the subunits Cdc10, Cdc3, Cdc12 and Cdc11 or Shs1 has already been purified as recombinant protein complex from crude *E. coli* lysates[13,15,23]. Previous work indicated that the flexible or unstructured N- and C-terminal extensions impede crystallization[20,21]. We removed the unstructured extensions from all four subunits and coexpressed Shs1$_{G21-S339}$-Cdc12$_{M1-G314}$-Cdc3$_{Q81-A410}$-Cdc10$_{G30-R322}$ together in *E. coli*. The resulting complex could be readily purified in sufficient purity and yield for crystallization trials (Supplementary Fig. 1A). MS analysis proved the nearly stoichiometric occurrence of all four subunits in the preparation (Supplementary Fig. 1B).

Diffracting crystals of the complex belonged to the space group C-1-2-1 and contained a tetramer in the asymmetric unit composed of the expected[13,24] Shs1-Cdc12-Cdc3-Cdc10 arrangement. The structure was solved at a resolution of 3.24 Å by molecular replacement using AlphaFold[25] models (Fig. 1A, Table 1) and submitted to the PDB under the identifier 8PFH. G-interfaces are formed between Shs1 and Cdc12 as well as between Cdc3 and Cdc10. Cdc12 and Cdc3 are connected by a NC-interface. Representative electron density figures of these interfaces are shown in Supplementary Fig. 2.

A homodimeric Cdc10 NC-interface is not present in the crystal and cannot be generated by symmetry operations. However, a filament-like structure with the subunit arrangement Shs1-Cdc12-Cdc3-Cdc10:Shs1-Cdc12-Cdc3-Cdc10 can be generated, indicating that the crystallization conditions favored the formation of a non-native Cdc10-Shs1 interface (Fig. 1A). All septin subunits within the complex consist of the common modified Rossmann fold. The repetitive αβ-units fold into a central β-sheet consisting of five to six β-strands, surrounded by seven α-helices plus the additional N-terminal α0-helix[8,16]. β6 is often only composed of a few amino acids and thus is not always annotated as strand. The main structural features of a typical septin subunit are highlighted in Fig. 1B.

All subunits contain GDP without a coordinated Mg$^{2+}$ ion (Fig. 2A). We confirmed the nucleotide content of the Shs1-tetramer by denaturing the purified septin complex and quantifying the released nucleotides on an analytical ion exchange column. The subunits in our septin complex preparation contained about 90% GDP and only trace amounts of GTP (Fig.2B, C).

### Features of the GDP bound G-interfaces in yeast septins.

We describe next the key features of the G-interfaces and nucleotide binding pockets in our crystal structure and follow therein the recently established nomenclature "residue(motif)"[8].

The structures of the SUEs with the distinct β-meander are solved for all subunits (Fig. 1) and replicate the conserved interactions known from human septins[8].

The switch 2 loops of all subunits are well structured. Switch 1 is fully resolved for Cdc10 and partly resolved for the remaining subunits.

The P-loop consensus sequence of the four subunits in our structure is G$_1$-L/T/I$_2$-G$_3$-K$_4$-T/S$_5$-T/A$_6$ (Fig. 1C). Four residues from the P-loop are in contact with the β-phosphate of the GDP (Fig. 3A). G$_1$, K$_4$ and T/S$_5$ make contact with the phosphate in all four subunits. L$_2$ and T$_2$ make the fourth contact in Cdc10 and Shs1, respectively, whereas G$_3$ makes the contact in Shs1 and Cdc3. The residues in position 6 coordinate the GDP α-phosphate in all four subunits (Fig. 3A).

Two residues from the switch 1 loop of Shs1 (Shs1$_{H51}$, Shs1$_{K52}$) make further contacts to the α-phosphate and β-phosphate,

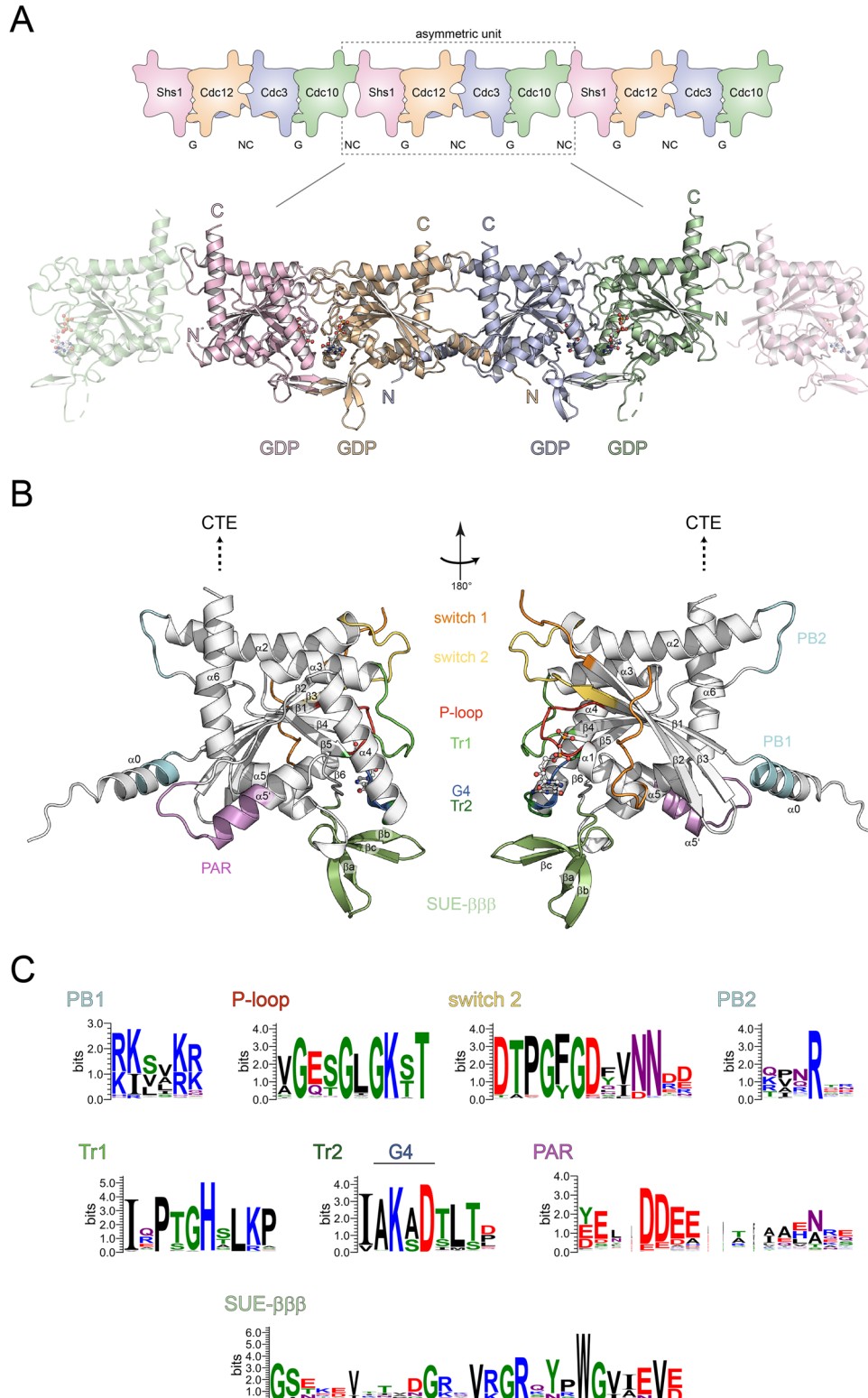

**Fig. 1 Structure of a tetrameric septin complex. A** Crystal structure of the yeast septin complex with a tetramer in the asymmetric unit. The interfaces are indicated and the nucleotides are displayed in ball-and-stick presentation. Filaments with non-physiological Shs1-Cdc10 interfaces are present in the crystal and can be generated by symmetry operations. **B** Subunit Cdc3 with annotated structural key features. Sheets and helices are numbered according to the classical G-domain numbering of Ras-like proteins. SUE: septin unique element; Tr: trans-loop; PB: poly-basic motif; PAR: poly-acidic region. The GDP is shown as ball-and-stick presentation. **C** Consensus sequences of selected structural features, created with WebLogo[50]. Alignments were fetched from Phylome-DB[51].

**Table 1 Crystallographic data collection and refinement statistics.**

| | Shs1-Cdc12-Cdc3-Cdc10 |
|---|---|
| **Data collection** | |
| Space group | C 1 2 1 |
| Cell dimensions | |
| $a, b, c$ (Å) | 258.178, 69.373, 92.533 |
| $\alpha, \beta, \gamma$ (°) | 90, 109.68, 90 |
| Resolution (Å) | 121.55–3.24 (3.65–3.24)[a] |
| $R_{merge}$ | 0.521 (1.395) |
| $I / \sigma I$ | 3.2 (1.5) |
| Completeness (%) | 63.2 (10.7)[b] |
| Redundancy | 6.6 (6.9) |
| **Refinement** | |
| Resolution (Å) | 121.55–3.24 (3.65–3.24) |
| No. reflections | 15692 (82) |
| $R_{work}/R_{free}$ | 0.282/0.290 |
| No. atoms | 9086 |
| Protein | 8974 |
| Ligand/ion | 112 |
| Water | 0 |
| B-factors | |
| Protein | 48.88 |
| Ligand/ion | 42.35 |
| R.m.s. deviations | |
| Bond lengths (Å) | 0.003 |
| Bond angles (°) | 0.519 |

A single crystal was used for the structure. Values in parentheses are for the highest-resolution shell.
[a]High-resolution reflexes were included according to published criteria[48,49].
[b]The traditional spherical completeness is given. The ellipsoidal completeness from the anisotropic analysis by autoPROC Staraniso[41] is 88.0 (36.4) with diffraction limits of 3.24 Å, 3.42 Å and 4.33 Å along the reciprocal axes (0.934 a* − 0.358 c*), b* and (0.954 a* + 0.299 c*), respectively.

respectively (Fig. 3B). $Shs1_{K36}$ and $Shs1_{K52}$ are placed in a triangular arrangement with the β-phosphate, $Shs1_{K36}$-$P_β$-$Shs1_{K52}$. The conserved presumably catalytic Thr(Sw1) is in Shs1 replaced by a glycine ($Shs1_{G102}$) which is not in contact with the nucleotide. The first position of the otherwise strictly conserved DXXG motif[26] in the switch 2 loop of Shs1 is replaced by a methionine. The switch 1 loop of Shs1 is longer than in the other canonical septins and is – except $Shs1_{H51}$ and $Shs1_{K52}$ - not in contact with the nucleotide. Instead, it snuggles along the surface of the Shs1 subunit where it folds into an interrupted β-strand between V68 and N80 which is attached to the β2-strand of the central β-sheet (Fig. 3C).

A histidine from Cdc12 ($Cdc12_{H155}$) is in contact with the GDP α-phosphate in Shs1, forming another triangular arrangement $Cdc12_{H155}$-$P_α$-$Shs1_{H51}$ (Fig. 3B). $Cdc12_{H155}$ belongs to the conserved Tr1 loop located between the β4-strand and the α3-helix. Analogous inter-subunit contacts of His(Tr1) with the nucleotide of the neighboring subunit are maintained by $Shs1_{H193}$ with the GDP bound in Cdc12 and $Cdc3_{H262}$ with the GDP bound in Cdc10.

Cdc10 displays a lysine ($Cdc10_{K155}$) instead of the phosphate-contacting histidine. $Cdc10_{K155}$ is, however, not in contact with the GDP of its neighboring subunit Cdc3.

Cdc3 also lacks the catalytic Thr(Sw1) which is replaced by a lysine ($Cdc3_{K181}$). Switch 1 of Cdc3 is mostly oriented perpendicular to the rod axis and does not participate in any nucleotide interaction.

The presumably catalytical residues in Cdc10 and Cdc12, Thr(Sw1) and Gly(DXXG), do not make interactions with the GDP in these subunits.

All septins possess a highly conserved arginine in the SUE β-meander (Arg(βb)), representing the central element of an overall conserved interaction hub in the nucleotide-binding pocket. Arg(βb) is coordinated by Asp(G4) and sandwiches together with Lys(G4) (Arg(G4) in Shs1) the guanine ring in a π-cation stack (Fig. 3D). It occupies the position taken otherwise by a lysine of the G5 motif (e.g. Ran:GDP, PDB-ID 5CIT) or alternatively a phenylalanine in switch 1 (e.g. KRas:GDP, PDB-ID 6MBU) of Ras-type GTPases. These residues normally coordinate the guanine ring with the positive charge or a T-shaped π-stack, respectively. Arg(βb) of the septins connects the G4 motif to an absolutely conserved Glu(α4) of the neighboring subunit, thereby maintaining a stabilizing inter-subunit contact (Fig. 3D). This inter-subunit contact is additionally maintained by a hydrogen bond between the amino group of the guanine ring and the backbone oxygen of a residue from the Tr2 loop of the neighboring subunit located five residues upstream of Glu(α4) (Ser/Ile from Shs1 or Cdc3, Thr from Cdc10 or Cdc12).

**AlphaFold reconstitution of octameric yeast septin complexes.** The introduced truncations of the septin subunits allowed the crystallization of the tetrameric structure but interfered with the formation of the native octamer. We tried the AlphaFold-based algorithm optimized for multimeric target structures[22] to predict the elusive Cdc10-Cdc10 interface that connects two tetramers to the palindromic octamer.

AlphaFold correctly assembled all G- and NC-interfaces within the crystallized tetramer from the predicted structures of dimers (Shs1-Cdc12, Cdc12-Cdc3, Cdc3-Cdc10).

The predictions were extremely accurate (RMSD-Cα values between 1.2 Å and 1.4 Å; Supplementary Fig. 3 and Supplementary Table 1), suggesting that a prediction of the missing Cdc10-Cdc10 dimer would be equally correct. The predicted Cdc10-Cdc10 dimer displays a NC-interface with anchored α0-helices (Fig. 4A) very similar to the Cdc12-Cdc3 NC-interface. PDB files of the predicted Cdc10 dimer and all other AlphaFold predictions are provided as Supplementary Data 1. We chose a crosslinking approach to experimentally validate the accuracy of the AlphaFold prediction of the Cdc10 dimer. If the algorithm places residues correctly within the interface, they should become available for crosslinking provided they are in a suitable distance and geometry. Exchanging residues in close distance against cysteines would allow to test their proximity across the interface by the degree of disulfide bond formation. We employed another machine learning-based algorithm[27] to predict residues in the $Cdc10^A$-$Cdc10^B$ NC-interface suitable for disulfide bond engineering. We selected three prominent contact sites within the interface with high confidence scores (Supplementary Fig. 4): One contact site between the two α0-helices (residues $Cdc10^A_{T19}$ and $Cdc10^B_{E23}$), one contact site between the α0-helix and the α6-helix of the neighboring subunit (residues $Cdc10^A_{Q21}$ and $Cdc10^B_{Q281}$) and one contact site between the PB2 motif and the α6-helix of the neighboring subunit (residues $Cdc10^A_{A132}$ and $Cdc10^B_{L300}$) (Fig. 4A and Supplementary Fig. 4). We mutated each residue of the native, non-truncated Cdc10 subunit individually or in combination to cysteine (disulfide bridge mutants), yielding two identical disulfide bridges in the octameric context.

All mutated Cdc10 subunits were co-expressed with the other full-length subunits in an *E. coli* strain enabling cytoplasmic disulfide bond formation. The septin complexes were purified by IMAC under non-reducing conditions and analyzed subsequently via Western blot under both reducing and non-reducing conditions. While the single mutants were always detected at

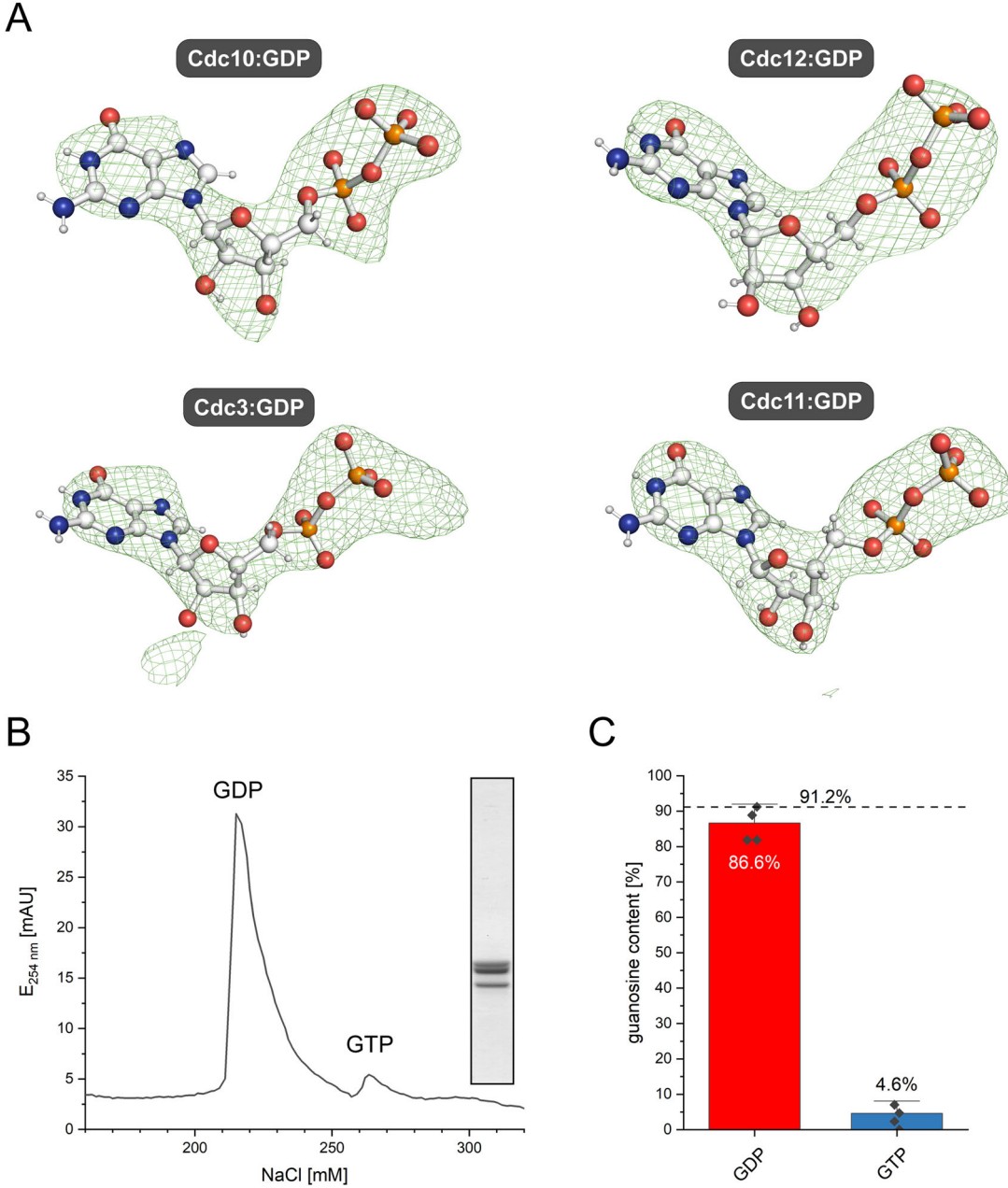

**Fig. 2 Nucleotide content of the crystallized tetramer. A** Polder-OMIT maps (contoured at 4.0 σ) show that all subunits contain GDP. **B** The chromatogram of a representative analytical IEX chromatography reveals the presence of mainly GDP. The inset shows a Coomassie stained SDS-PAGE gel of the purified complex which was subsequently denatured for nucleotide content determination. **C** Integration of the peaks from the IEX chromatograms (N = 4, error bars represent SD) confirms that the complex contains 87% GDP. The remaining GTP originates likely from a mixed population in one of the subunits.

the running position of the Cdc10 monomer, a considerable fraction of the disulfide bridge mutants was shifted to the expected running position of a dimeric Cdc10 under non-reducing conditions (Fig. 4B). These experiments indicate successful disulfide bridge crosslinking and thus correct positioning of the residues by the AlphaFold algorithm.

We aligned the predicted and experimentally validated Cdc10-Cdc10 dimer to our crystal structure, yielding a model for a yeast septin octamer (Fig. 4C).

During the preparation of this manuscript, the crystal structure of a Cdc3-10-10-3 tetramer became available (PDB-ID 8SGD)[28]. This structure covers the central Cdc10-Cdc10 interface. Aligning

the experimental interface with our predicted Cdc10 homodimer reveals a RMSD-Cα as low as 1.25 Å (Supplementary Fig. 5). This extremely good agreement confirms our approach as well as the predicted structure of the septin octamer and emphasizes the remarkable performance of the AlphaFold algorithm.

Is AlphaFold also able to predict larger septin assemblies? We tried the algorithm by assembling an octameric rod from two copies of the sequence of each subunit including here the terminal subunit Cdc11. We learned that the algorithm was not able to suggest a meaningful structure of the octamer, but when given only one copy of each sequence, the tetramer was assembled in the correct order (Supplementary Fig. 3). The Cdc12-Cdc3-Cdc10

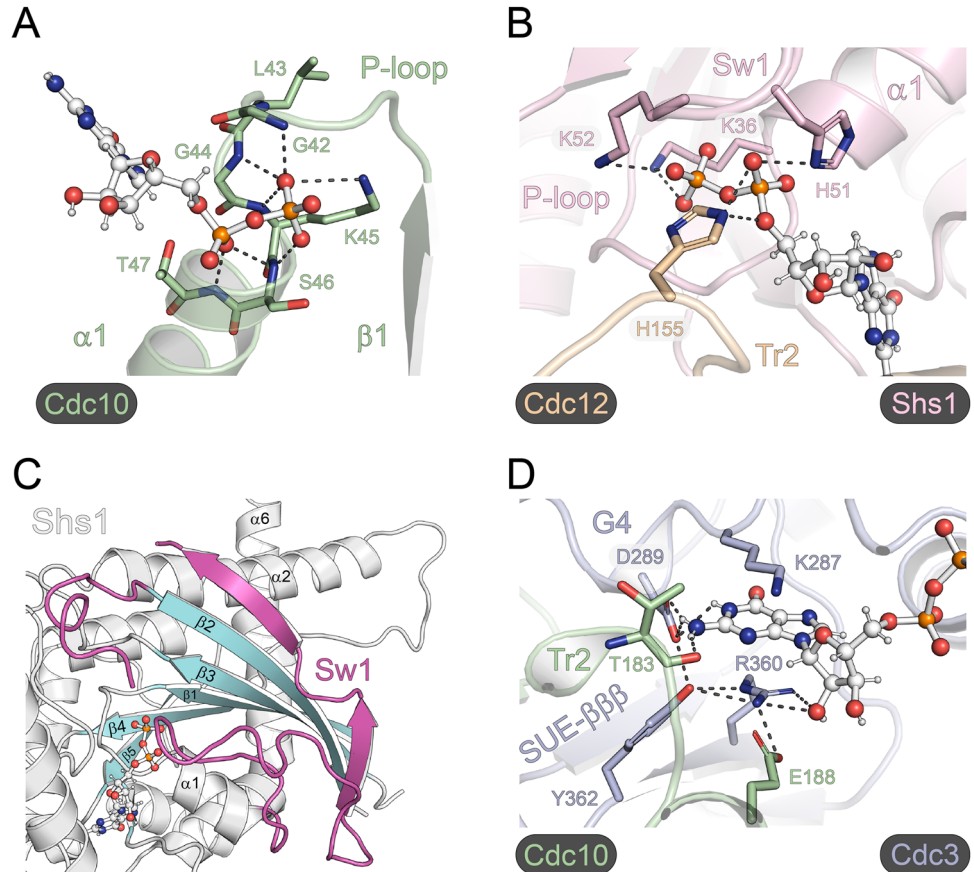

**Fig. 3 Features of the G-interfaces. A** Coordination of the β-phosphate by residues from the P-loop, shown exemplarily for Cdc10. **B** Triangular arrangement by Shs1$_{K36}$ and Shs1$_{K52}$ with the β-phosphate. Another triangular arrangement is formed at the α-phosphate in Shs1 by Shs1$_{H51}$ and Cdc12$_{H155}$ from the neighboring subunit. **C** The exceptionally long switch 1 loop (colored magenta) of Shs1 snuggling along the surface of the Shs1 subunit. **D** The highly conserved Arg(βb) is the central element of a conserved interaction hub in the nucleotide-binding pocket, shown here for Cdc3$_{R360}$. Arg(βb) is coordinated by Cdc3$_{D289}$ and sandwiches together with Cdc3$_{K287}$ the guanine ring in a π-cation stack. Arg(βb) connects furthermore the G4 motif to a conserved Glu(α4) of the neighboring subunit (here Cdc10$_{E188}$), maintaining a stabilizing inter-domain contact. Dotted lines in A, B and C indicate hydrogen bonds.

subunits from this predicted complex showed a RMSD-Cα of only 2.34 Å compared with the respective subunits from our crystal structure (Supplementary Table 1).

**A hydrophobic element stabilizes the NC-interface in septin complexes.** The NC-interface of septins is composed of the α0-, α2- and α6-helices and the loop connecting β2 and β3[7,8,19]. In the Cdc3-Cdc12 NC-interface, the loops connecting α2 and β4 form a distinct salt bridge network together with a conserved glutamate and arginine residue of the α6-helix in the upper part of the interface. These features were already described for human septin NC-interfaces[19] (Fig. 5). The lower part of the interface is stabilized by interactions of six hydrophobic residues, ranging from the hook loop preceding the α0-helix over the helix itself to a conserved Phe(β1). These residues are positioned as a crest which inserts into a hydrophobic cleft of the neighboring subunit (Fig. 5). This cleft is formed by hydrophobic residues present in the β1-, β2- and β3-strand as well as in the loop connecting β2 and β3 and the region connecting the C-terminal and N-terminal parts of α5 and α6, respectively (Fig. 6A). The otherwise conserved Phe(β1) is in Cdc12 replaced by a glycine (Cdc12$_{G33}$), but in the three-dimensional fold the position is filled by Cdc12$_{F90}$ of the loop connecting the β2- and β3-strand, thus making the phenylalanine a spatially conserved residue.

Analysis of the NC-interface forming sequences of yeast and human septins revealed that the hydrophobic crest is highly conserved and embedded in a general 19 residue motif with the sequence **V/I/F$_1$**-G$_2$-**F/I$_3$**-X$_4$-X$_5$-**L/I/V$_6$**-P/h$_7$-X$_8$-Q$_9$-**h$_{10}$**-X$_{11}$-X$_{12}$-X$_{13}$-X$_{14}$-**h$_{15}$**-X$_{16}$-X$_{17}$-X$_{18}$-**F/I$_{19}$** (h - any hydrophobic amino acid; bold printed - conserved key positions; X – any amino acid) (Fig. 6B). Position 10 is the least conserved position of the motif. Cdc12 and Spr3 possess an arginine at this position which is anchored by two glutamate residues (Cdc3$_{E390}$ and Cdc3$_{E394}$) from the α6-helix in the neighboring Cdc3 (Fig. 5, lower panel).

Cdc10 lacks the α0-helix in our expression construct, explaining the tetrameric nature of the complex. We asked if addition of the α0-helix containing the hydrophobic crest to Cdc10 is sufficient to restore an octamer. The resulting Shs1$_{G21-S339}$-Cdc12$_{M1-G314}$-Cdc3$_{Q81-A410}$-Cdc10$_{M1-R322}$ construct eluted at about the same retention volume from an analytical SEC than an untruncated octamer and considerably earlier than the tetrameric construct (Fig. 7A, Supplementary Fig. 6). The Cdc10 α0-helix is consequently sufficient and necessary for octamer formation.

Having validated the role of the α0-helix including the hydrophobic crest for NC-interface and octamer formation, we attempted to better characterize the forces that drive the formation of this newly described element. We calculated the relative solvent accessible surface area (SASA) of each residue in the Cdc10-Cdc10

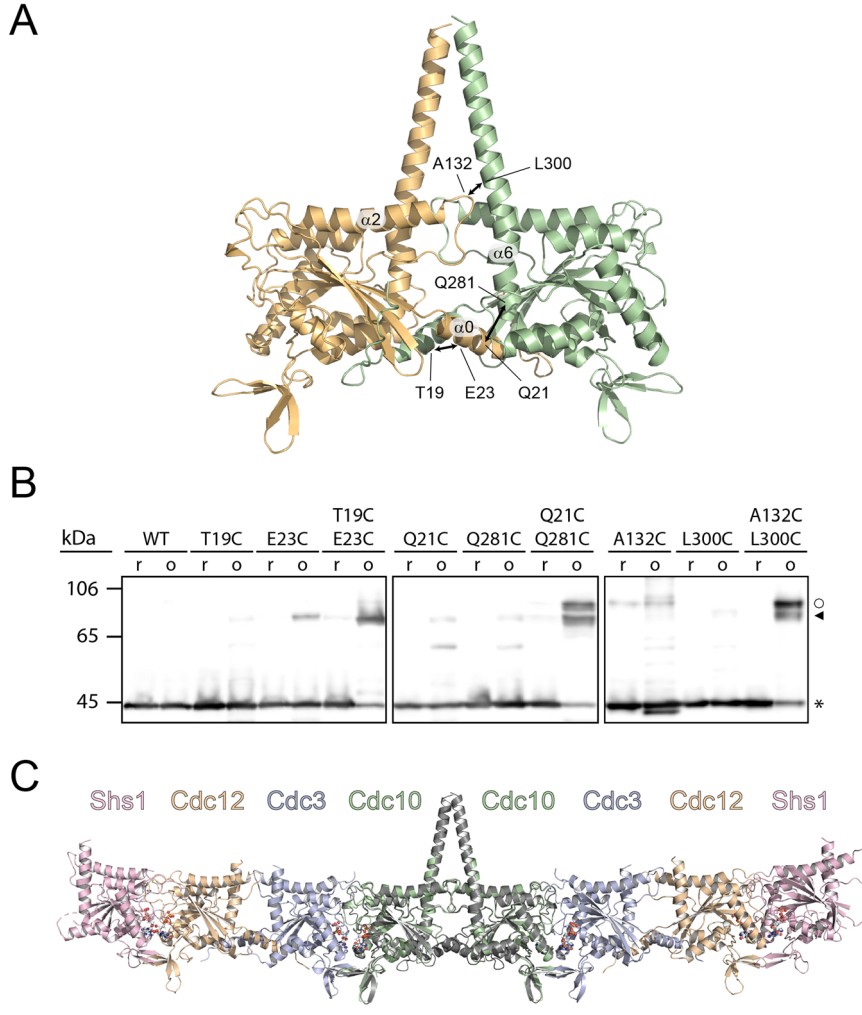

**Fig. 4 AlphaFold assisted assembly of the yeast octameric rod. A** Structural model of the predicted Cdc10-Cdc10 NC-interface. Positions of the residues selected for disulfide bond engineering are indicated. **B** Disulfide bond engineering leads to crosslinking in oxidizing (o) but not in reducing (r) conditions. Double (marked with an arrowhead) and single (marked with a circle) crosslinks entail different packing density of the backbone chains for the Q21C/Q281C and A132C/L300C mutants. Non-crosslinked Cdc10 is marked with an asterisk. Shown is a Western blot detecting the S-tag fused to Cdc10 in an octameric septin rod preparation. **C** Structural model of an Shs1 containing octameric septin rod assembled from the tetrameric crystal structure and the predicted Cdc10-Cdc10 interface (dark-gray).

NC interface and found that the six conserved residues at positions 1, 3, 6, 10 and 19 of the hydrophobic crest motif have relative SASA values of 1-7% indicative of being deeply buried within the interface. Only position 15 has a SASA value of 14% (Fig. 6A). The SASA value should inversely correlate with the strength of the hydrophobic stabilization derived from these residues. To confirm our prediction, we mutated all conserved hydrophobic residues of the hydrophobic crest in Cdc10 (V13, F15, I18, I22, L27 and F31; corresponding to the key residues at positions 1, 3, 6, 10, 15 and 19 in the hydrophobic crest motif outlined above) individually to alanine.

We purified the mutated septin complexes and subjected them to analytical SEC. Compared to wildtype, all mutants except $Cdc10_{L27A}$ eluted as tetramer (Fig. 7B, the reference chromatograms for the tetrameric and octameric constructs are shown in Supplementary Fig. 6), demonstrating that these mutations disrupted the native NC-interface. The $Cdc10_{L27A}$ mutant at position 15 of the hydrophobic crest motif has the highest SASA value (14%) and eluted in a single peak migrating between the tetramer and octamer peaks (Fig. 7B). This elution profile correlates well with an already described equilibrium between octamer and tetramer[29].

Replacing the non-conserved $Cdc10_{L26}$ (corresponding to position 14 in the hydrophobic crest motif; solvent exposure 19%) with alanine did not interfere with NC-interface integrity as the mutant septin complex eluted as the wildtype octamer (Fig. 7B).

We next asked whether the mutations in the hydrophobic crest disrupt essential features of the septins in the context of the living cell. We generated a Cdc10 knockout yeast strain by replacing the *CDC10* gene with an antibiotic cassette. The essential functions of Cdc10 were provided by a centromeric rescue plasmid containing beside the unaltered *CDC10* an Ura3 prototrophy. Hydrophobic crest mutants in Cdc10 and suitable controls were subsequently introduced into this strain via a second centromeric plasmid. Yeast cultures bearing both plasmids were spotted onto FOA plates, forcing the cells to kick out the rescue plasmid. Growth or non-growth on FOA-containing media should indicate whether the introduced mutations interfere with the essential functions of Cdc10. Deleting the entire α0-helix of Cdc10 abolished the growth of the corresponding strain on FOA media thus confirming the suitability of the test system (Fig. 7C).

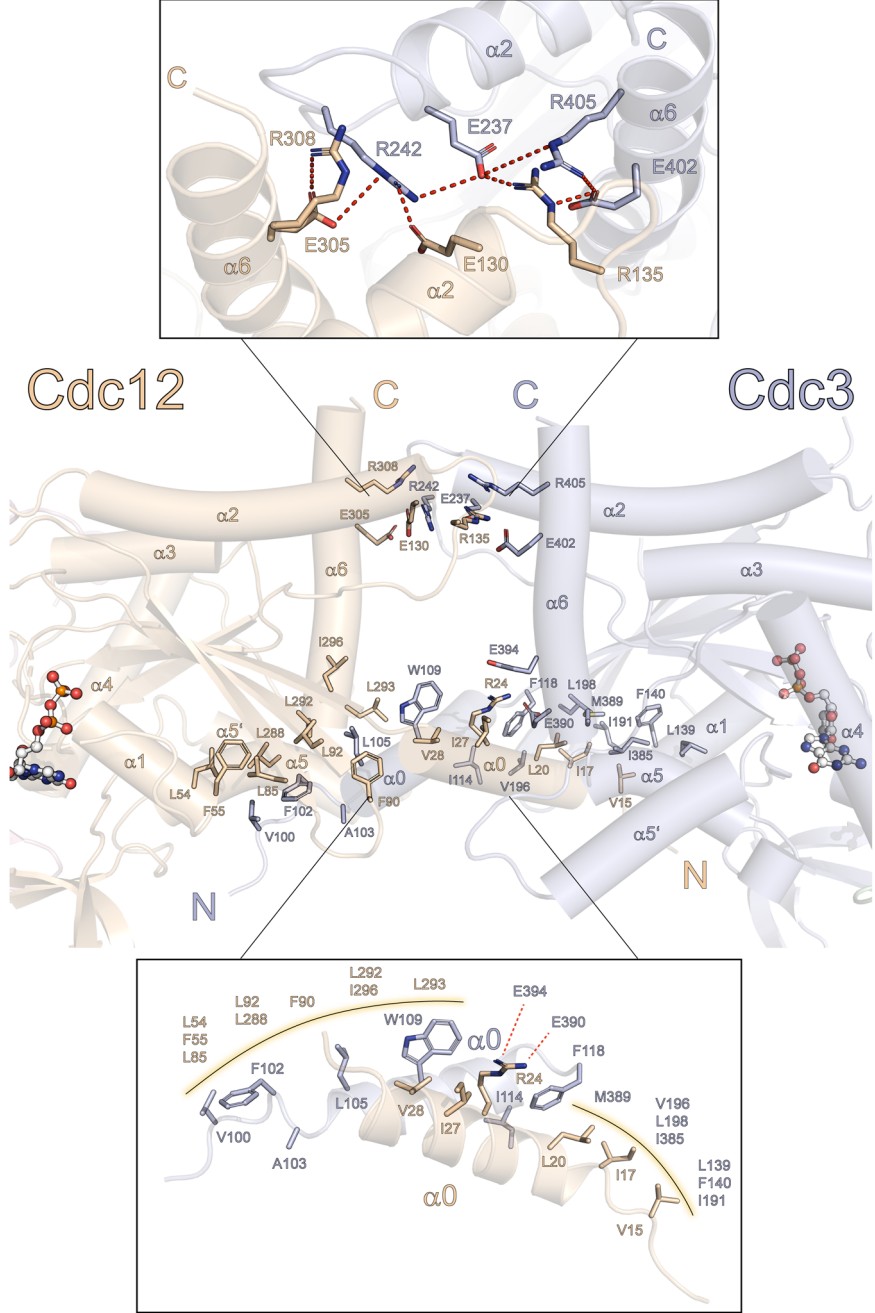

**Fig. 5 Features of the Cdc12-Cdc3 NC-interface.** A salt bridge network stabilizes the upper part of the interface (upper panel). Charged interactions are indicated by dotted lines. Hydrophobic interactions stabilize the lower part of the interface (lower panel) maintained by hydrophobic residues arranged like a crest, ranging from the hook loop over the α0-helix to the conserved Phe(β1) (here Cdc3$_{F118}$ and Cdc12$_{G33}$, structurally replenished by Cdc12$_{F90}$).

The F15A and I18A mutations in Cdc10 were lethal for the cells, pointing towards the significance of these positions for octamer formation. Growth of cells bearing the V13A mutation was severely impeded whereas the I22A and F31 mutation had a mild, but still clearly detectable growth defect. The growth of the strains bearing the L26A and L27A mutations was indistinguishable from the wild type (Fig. 7C).

To substantiate this finding, we evaluated the contribution of each introduced mutations on the stability of the interface using the software FoldX[30]. FoldX predicts changes in interaction energies upon point mutation in a provided structure. The effect of the mutations on the predicted ΔΔG values correlate well with the observed phenotypes (Fig. 7C).

As a further control for our structural interpretation of the hydrophobic crest, we introduced the lethal I18 A mutation in the disulfide bridge mutant Cdc10$^A_{A132}$-Cdc10$^B_{L300}$ which introduces an artificial disulfide bridge between the PB2 motif and the α6-helix of the neighboring Cdc10 subunits (see above). The I18A mutation entirely prevented the otherwise successful disulfide crosslink whereas the L27A and the non-conserved L26A mutant did not impede crosslinking (Fig. 7D). Both experiments strongly confirm our characterization of the hydrophobic crest residues.

**The hydrophobic crest is a species-overlapping structural element in septins.** Our sequence analysis showed that the hydrophobic crest is a conserved structural feature in yeast and

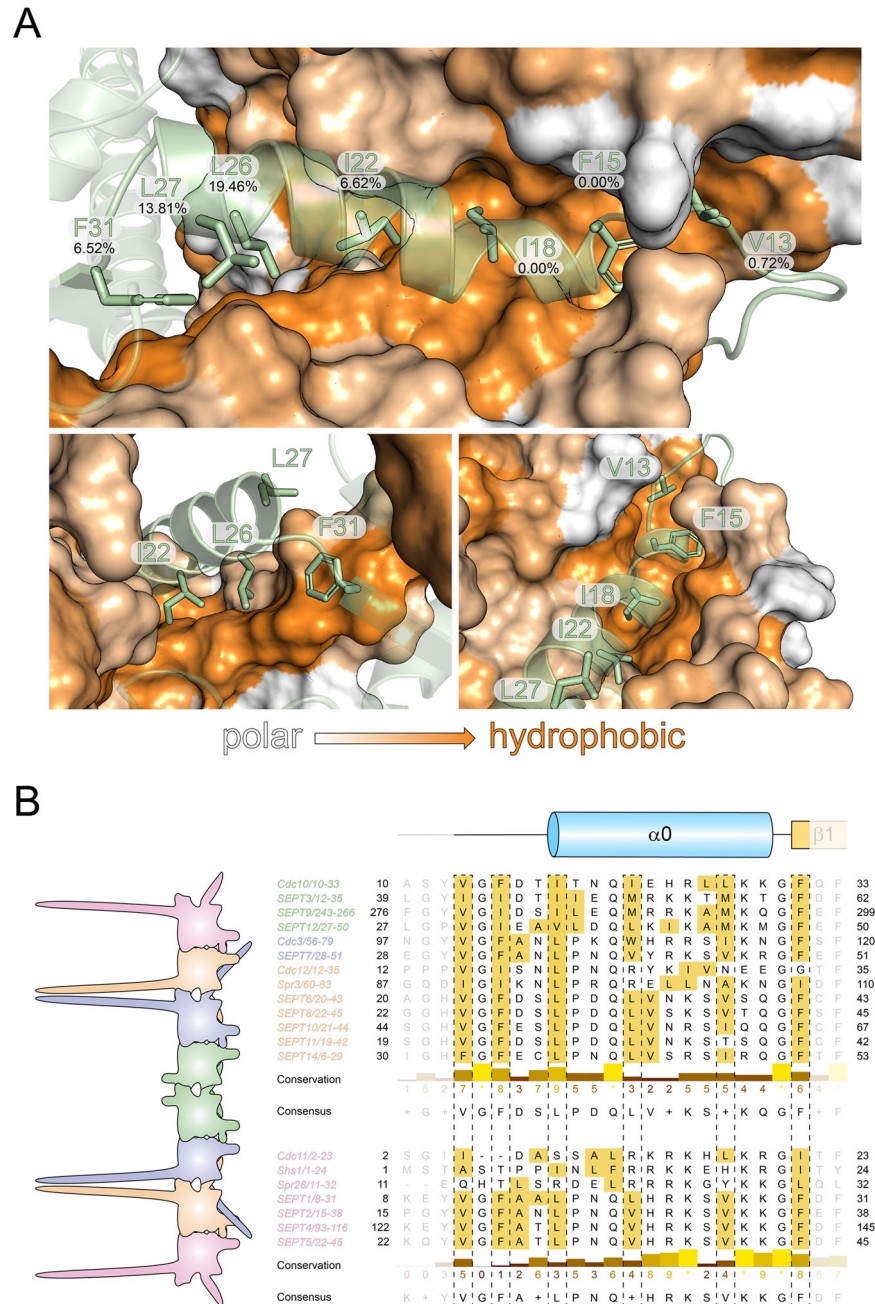

**Fig. 6 The hydrophobic crest is a conserved structural element. A** Molecular surface representation of one subunit of the predicted Cdc10-Cdc10 NC-dimer, colored by residue hydrophobicity from white (polar) to orange (hydrophobic) according to the Eisenberg normalized consensus hydrophobicity scale. The other Cdc10 subunit is represented as ribbon with the sidechains of the hydrophobic crest shown. The relative solvent accessible surface area (SASA) of each residue is provided. **B** Sequence alignment of yeast and human septins (ordered by their position in the rod) shows that the hydrophobic crest is a conserved 19 residue-motif. The conservation degree (10: absolutely conserved, 0: not conserved) and the consensus sequence is provided. Residues with a hydrophobicity score >0.5 according to the Eisenberg scale are colored yellow-orange.

mammalian septins (Fig. 6B). Structures of the physiological NC-interfaces in the human septins SEPT6/7 and SEPT2/2 clearly reveal the essential features of the hydrophobic crest (Fig. 8A). Another NC-interface is formed between two SEPT9 subunits. The structure of this interface is only available from crystallographic symmetry operations and does not show the α0-helix[31]. SEPT9 is—analog to Cdc10—the central dimer of the canonical human octameric septin rod[11] and thus particularly suited to test the significance of the hydrophobic crest for NC-interface formation. We selected the conserved residue SEPT9$_{F297}$ and the weakly conserved SEPT9$_{M288}$, corresponding to the positions 19

and 10 of the hydrophobic crest motif outlined above. We mutated both residues individually to alanine and subjected the purified mutated human septin complexes to analytical SEC. The wild-type complex (with the unstructured N-terminal extension of SEPT9 removed) eluted with the retention volume of the octameric complex. Removal of the entire α0-helix from SEPT9 resulted in a complex eluting as tetramer (Fig. 8B).

SEPT9$_{F297A}$ eluted in a single peak as tetramer. The SEPT9$_{M288A}$ containing complex construct eluted as a double peak with the retention volumes corresponding to a tetramer and an octamer (Fig. 8C). The elution profile suggests that the M288A

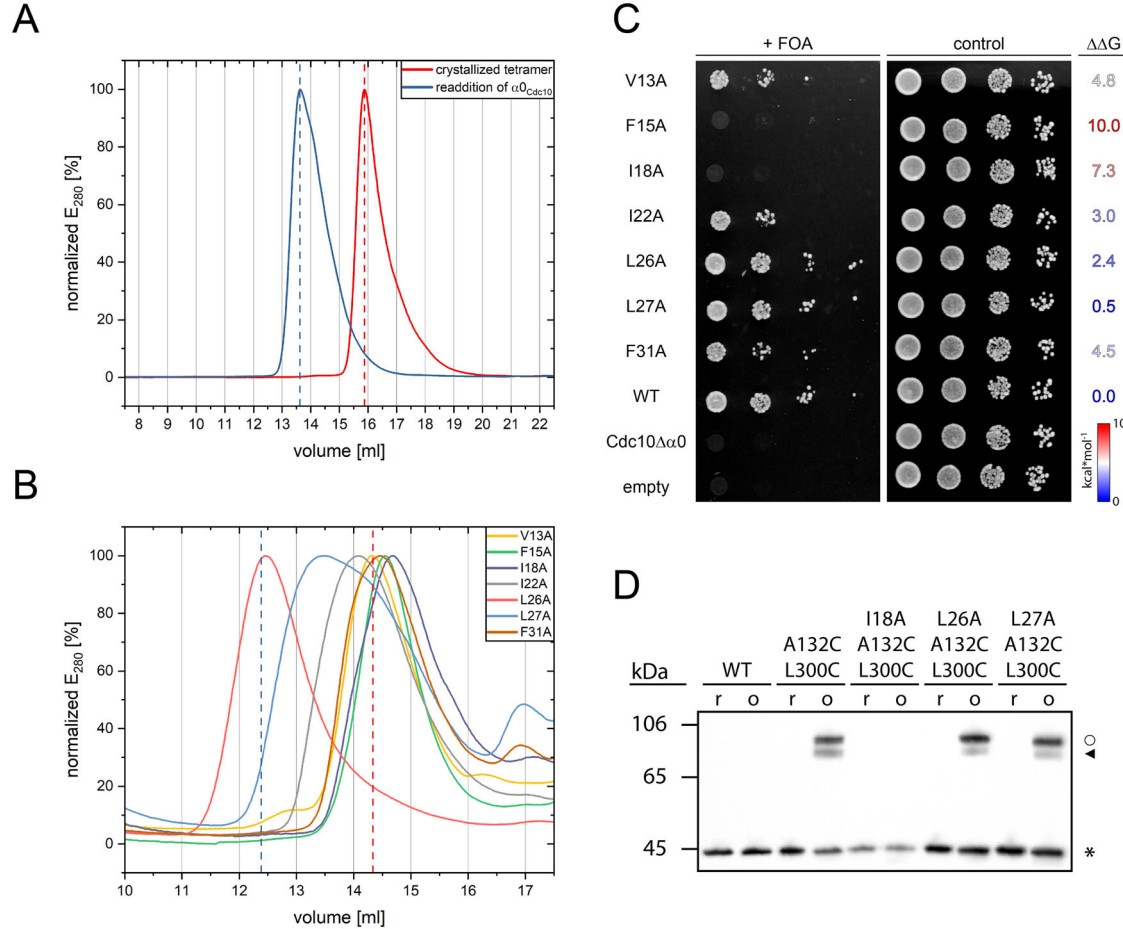

**Fig. 7 The hydrophobic crest is essential for NC-interface integrity in yeast septins. A** Addition of the α0-helix to the Cdc10 subunit of the truncated tetramer employed for crystallization is sufficient to restore an octamer. Shown are analytical size exclusion runs with normalized peak intensity.
**B** Mutations of the conserved hydrophobic crest positions in Cdc10 interrupt NC-interface integrity leading to a tetrameric elution profile (indicated by the red dashed line). Mutation at the non-conserved L26 retains an octameric elution profile (indicated by the blue dashed line). The reference chromatograms for tetramer and octamer (dashed lines) are shown in Supplementary Fig. 6. Shown are analytical size exclusion runs with normalized peak intensity.
**C** Evaluation of hydrophobic crest mutants in Cdc10 in the context of the living yeast cell. Cdc10 mutants are expressed from a centromeric plasmid in a *CDC10* knock out strain after kick-out of a rescue plasmid on FOA medium. F15A, I18A and Δα0 mutants are lethal. V13A, I22A and F31A show a slight growth defect and L26A and L27A are indistinguishable from wildtype. ΔΔG values (kcal/mol) predicted by FoldX are provided. The higher the value, the higher the disruptive potential of the introduced mutation. **D** Introduction of the I18A mutation into a disulfide mutant in Cdc10 prevents successful crosslinking under oxidizing conditions whereas L26A and L27A do not interfere with crosslinking. Shown is a Western blot detecting the S-tag fused to Cdc10 in an octameric septin rod preparation. Labeling as in Fig. 4.

exchange readily weakens but not entirely interrupts the stability of the SEPT9-SEPT9 interface.

The effects of both mutations confirm that the hydrophobic crest is a conserved structural element present in septins from yeast and man.

## Discussion

The common building block of both human and yeast septins is an octameric rod. Low-resolution hexameric and trimeric structures of human septin complexes were solved or generated by symmetry operations from smaller building blocks[7,19], but structures with a resolution below 3.5 Å were so far elusive.

We obtained the structure of a septin tetramer from yeast septins at 3.2 Å. This structure lacks the central Cdc10-Cdc10 NC-interface that is required to form the octameric rod. We reconstituted this interface with AlphaFold and grafted it onto the tetramer structure, yielding a high-confidence model of the octameric septin rod.

Interestingly, the octamer assumes a slight banana-shape bending (Fig. 4C, Supplementary Fig. 5B) that was already reported for in vitro assembled human septin filaments[7] and observed in EM-imaged yeast septin octamers[7,13].

We substantiated the performance of the AlphaFold algorithm by utilizing another artificial intelligence-based tool[27] to identify residues in the Cdc10-Cdc10 interface that are close enough and in the right orientation for disulfide bridge engineering. All engineered contact sites crosslinked the two Cdc10 subunits in the octamer, confirming how experiment and artificial intelligence can complement each other to obtain novel structures.

Our approach was independently corroborated by the recently published experimental structure of the Cdc10-Cdc10 interface[28]. The root mean deviation (RMSD-Cα) between the predicted and experimentally solved interfaces was below 1.5 Å.

All subunits within the available multimeric septin structures from yeast and human contain a nucleotide whereas the monomeric Cdc11 crystal structure is nucleotide free[21]. In contrast to the terminal subunit Shs1 within the tetramer structure, the SUE-

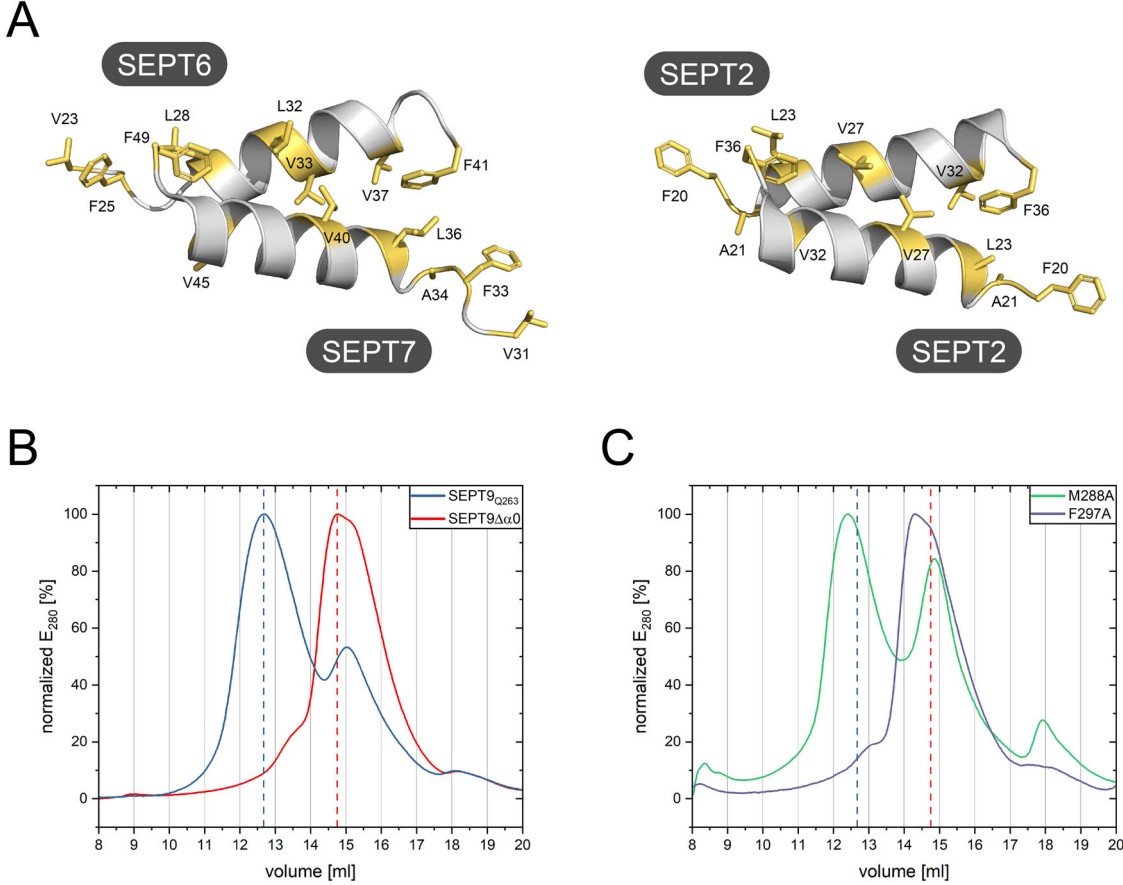

**Fig. 8 The hydrophobic crest is a species-overlapping feature in septins. A** Hydrophobic crest arrangements in the human septin interfaces formed between SEPT6 and SEPT7 (PDB-ID 7M6J) and the SEPT2 dimer (PDB-ID 2QA5). **B** Addition of the α0-helix (blue) to a truncated SEPT9 (red) restores a human SEPT2-6-7-9 containing octamer. Shown are analytical size exclusion runs with normalized peak intensity. **C** Introduction of a F297A mutant in the hydrophobic crest motif in SEPT9$_{Q263}$ (blue peak) results in a tetrameric elution profile (indicated by the red dashed line). Introduction of a mutation at the low-conserved M288 leads to a partial disruption of the hydrophobic crest, indicated by a double peak corresponding to an octameric and tetrameric elution profile, respectively (blue and red dashed lines). Shown are analytical size exclusion runs with normalized peak intensity. Reference profiles for octamer and tetramer (dashed lines) are from B.

βββ region is unresolved in the Cdc11-apo structure and the resolved anchoring residues suggest that the SUE-βββ is twisted away from the filament axis at a 90° angle (Fig. 9). We suspect that the conformational deviation of the SUE-βββ in the Cdc11-apo form results either from the missing G-interface partner and/or from the missing nucleotide. Recently an in silico study suggested that the strictly conserved Arg(βb) plays a pivotal role in septin G-interface dimerization[18]. Furthermore, it was shown that GDP is an interface-stabilizing factor. Arg(βb) coordinates the nucleotide in any subunit (Fig. 3D) and contributes thereby to interface stability. Since other septins are also apo proteins in their monomeric state (Cdc3[32], SEPT3[33], SEPT7[34]) we suggest that the nucleotide is indeed the factor that stabilizes septin G-interfaces by arranging the SUE via the interaction network around the Arg(βb).

The classic loaded spring mechanism of small GTPases predicts coordination of the γ-phosphate by a highly conserved glycine from the DXXG motif in switch 2. A magnesium ion is coordinated by a threonine from switch 1[26]. Shs1 and Cdc3 lack this catalytic threonine. Cdc11 (the alternative terminal subunit lacking also the catalytic threonine) and Cdc3 were furthermore predicted to be catalytically inactive based on experimental data[16,35]. Since we loaded the purified complex with GTP prior to crystallization, it was surprising to find the bona fide inactive Shs1 and Cdc3 subunits in a GDP bound state. We propose that Shs1

and Cdc3 have either a higher affinity for GDP impeding GTP uptake or that they hydrolyze GTP via another mechanism than the loaded spring mechanism. SEPT2 was shown to have similar affinities for GDP and the GTP analogon GppNHp in in vitro nucleotide exchange assays[20] but experimental data for yeast septins are entirely missing. Further experimental research is required to solve this conundrum.

Whereas the role of the nucleotide in G-interface integrity is still a matter of debate and ongoing research, the NC-interface received much less attention since only few structural data of low- or intermediate-resolution structures of native interfaces (i.e. not generated by symmetry operations) are available. However, especially the salt bridge network stabilizing the upper part of the interface, present also in our structure, is conserved and well characterized in human septins[7,19,31]. Current literature states that the α0-helices within the lower part of the NC-interface are anchored by two conserved hydrophobic residues (mostly Phe), one in the hook loop preceding the α0-helix and one in the following β1[8]. We identified additionally an array of six conserved hydrophobic residues within a 19-residue motif ranging from the hook loop to the first residue of β1 as indispensable part of the NC-interface. Mutation of any residue from this array, which we termed hydrophobic crest, resulted in disruption or at least distortion of the interface in vitro and in vivo. The only exception is the L27A mutant at position 15 in the hydrophobic

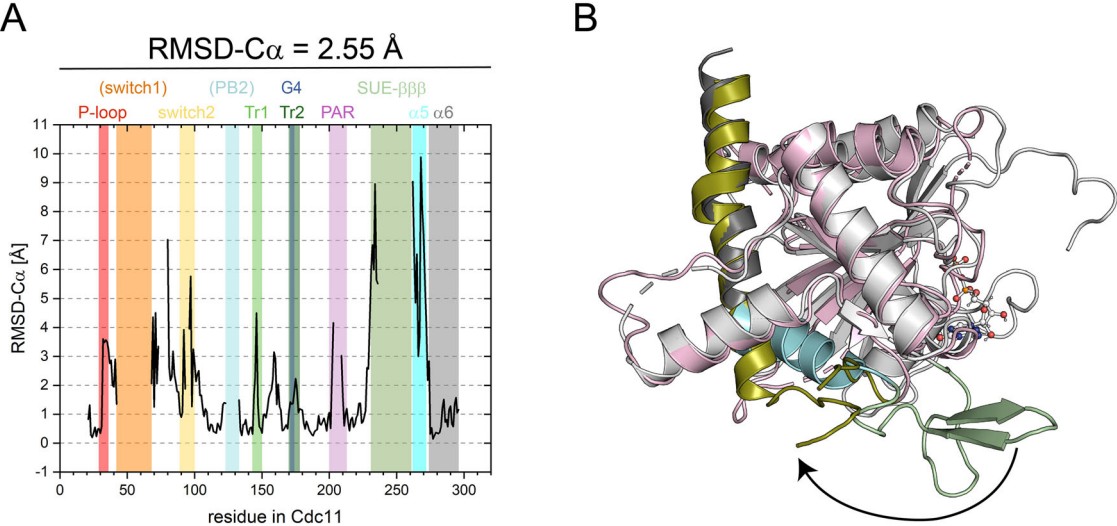

**Fig. 9 Comparison of the Cdc11-apo structure with Shs1. A** "Per residue" RMSD-Cα plot Cdc11-apo (PDB-ID 5AR1) vs. Shs1:GDP from the tetrameric crystal structure with highlighted structural features. Switch 1 and PB2 are written in parentheses as they are not resolved in the Cdc11 structure. **B** Overlay of Shs1:GDP (gray) with Cdc11-apo (light-pink). The SUE elements in both proteins are highlighted in pale-green, cyan and dark-gray (Shs1) or deep-olive (Cdc11-apo) to emphasize the large conformational change.

crest motif in Cdc10. The observation that some septins have a less-hydrophobic valine (i.e. the complete 2b group of mammalian septins[10]) or alanine (Spr3) in this position, explains the modest effect of this mutation in vitro and in vivo.

We showed that septin complexes lacking the α0-helix in the central subunits Cdc10 and SEPT9 elute as tetramer from analytical SEC and that the addition of the helix restores an octamer. Deletion of this helix from Cdc10 was lethal for yeast cells in our complementation assay. Removal of the α0-helix from SEPT3 led to disruption of the central SEPT3 NC-interface in recombinantly expressed human SEPT3 containing septin octamers[29]. These results are altogether in line with published findings showing that the α0-helix is essential for NC-interface dimerization[6,13,36]. To investigate whether features of the hydrophobic crest are not only sequentially but also functionally conserved in human septins, we introduced two mutations in the hydrophobic crest motif in SEPT9, the alternative central subunit of the canonical human octamer from the SEPT3 subgroup.

In accordance with our model both mutations interfered with the integrity of the human septin NC-interface. We conclude that proper orientation of the α0-helix with its hydrophobic crest is also required in human SEPT9.

For the SEPT3 subgroup (including Cdc10, SEPT3, SEPT9 and SEPT12)[10] a "closed" conformation of the NC-interface was predicted in which the α0-helix splays outwards from the filament instead of contributing to interface integrity[8,31]. However, the underlying SEPT3 construct was present as a monomer in solution and the prediction of the closed conformation was based on an interface generated via symmetry operations[31].

Our experimental results contradict the concept of open and closed NC-interfaces and we suggest that the closed conformation is not a physiologically occurring phenomenon but rather an atypical interface resulting from the high protein concentrations used for crystallization.

The hydrophobic crest motif is conserved among the NC-interfaces of yeast- and human septins (Fig. 6B), however, the overall hydrophobicity is not. It is thought that a Cdc11-Cdc11 or a Shs1-Shs1 NC-interface drives the formation of the yeast septin filament. The terminal subunits Cdc11 and Shs1 possess only three hydrophobic residues at the otherwise conserved key positions of the hydrophobic crest. Their human counterparts from

the SEPT2 group have six hydrophobic key residues. This finding might explain why yeast septin octamers form filaments in vitro only under salt conditions below 100 mM NaCl whereas the formation of human terminal NC-interfaces occur already at higher salt concentration[13,23,37]. We propose that the absence of a strong hydrophobic effect in yeast septins makes interface formation more dependent on the salt bridge network of the NC-interface.

Why should a destabilized terminal NC-interface be favored in yeast septins? Septins in yeast undergo a coordinated 90° reorientation during the cell cycle which might involve disassembly and reassembly of filaments[38]. Such a coordinated re-orientation is not observed in human cells. The lower hydrophobicity of the terminal NC-interfaces of the yeast septins might thus facilitate this yeast-specific process.

## Methods

**Plasmids and strains**. The ORFs of Cdc3 and Cdc10 or Cdc12 and Shs1 with and without N- and C-terminal truncations were inserted into the compatible bicistronic expression plasmids pACYC-Duet-1 and pETDuet-1, respectively. All mutants in Cdc10 were generated by SOE-PCR.

For the complementation assay, a wild-type copy of Cdc10 was expressed from a Ura3 prototrophy containing rescue plasmid in a *CDC10* knock out yeast strain.

Cdc10 with and without mutations was introduced by a centromeric plasmid and the rescue plasmid was driven out by plating the resulting strains on medium containing FOA.

More elaborated methods and a list of all generated plasmids (Supplementary Table 2) are provided in the Supplementary Methods.

**Protein purification**. All protein expressions were conducted in *E. coli* BL21DE3 except for the disulfide engineering mutants which were expressed in *E.coli* T7-SHuffle Express, enabling disulfide bond formation. The employed plasmid combinations are listed in Supplementary Table 3.

Protein expression and purification for crystallization was performed as described[23] with some modifications: Expression was carried out in 2.0 L SB medium supplemented with 2% v/v

Ethanol abs. at the time of induction with IPTG. The crude extract (usually 200 mL) was applied onto a 5 mL HisTrap HP column (Cytiva) and eluted with an imidazole step gradient[23] after extensive washing.

The product peak was collected, desalted using a PD10 column (Cytiva) and subjected to anion exchange chromatography using a high resolution 6 ml ResourceQ column (Cytiva)[23]. The protein concentration of the collected product peak was determined via a Bradford assay and the septin complex was subsequently incubated with a five-fold molar excess per subunit GTP for 1 h at room temperature in the presence of 5 mM EDTA. The reaction was quenched by the addition of 10 mM MgCl$_2$ and subjected to size exclusion chromatography (SEC) on a Superdex 10/200 column (Cytiva) with 25 mM Tris pH 8.0, 300 mM NaCl, 5 mM MgCl$_2$ as running buffer. Purity of the product peak was judged from Coomassie-stained SDS-PAGE gels. The product peak was pooled, concentrated, and subsequently used for crystallization. The final protein concentration was determined at 280 nm using a NanoDropND-1000 spectralphotometer (Peqlab) with calculated extinction coefficients. Mass spectrometry to confirm the integrity of the septin comlex in the preparation was performed as described in the Supplementary Methods.

Protein expression and purification of septins for analytical size exclusion chromatography was performed in a two-step setup employing IMAC and IEX as described[23]. Analytical SEC was subsequently performed on a Superose 6 column (Cytiva) with 25 mM Tris pH 8.0, 300 mM NaCl as mobile phase.

Human septin complexes were purified by IMAC as described[37]. As second purification step anion exchange chromatography on a Resource Q column using Tris pH 8.0 as buffer system was performed. Analytical SEC was performed with 25 mM Tris pH 8.0, 500 mM NaCl as mobile phase.

The performance of the Superose 6 column was routinely assayed using a gel filtration standard (BioRad).

Disulfide mutants were expressed in 100 ml SB as outlined above and purification was performed by IMAC on a 1 ml HisTrap HP column (Cytiva) using non-reducing buffers for lysis and chromatographic purification (IMAC A: 50 mM Tris pH 8.0, 300 mM NaCl, 2 mM MgCl$_2$, 12% v/v Glycerol, 15 mM Imidazole. IMAC-B: 50 mM Tris pH 8.0, 500 mM NaCl, 2 mM MgCl$_2$, 12% v/v Glycerol, 500 mM Imidazole). After purification, protein concentrations were determined via Bradford assay and adjusted to 0.15 mg/ml. In the following, proteins were denatured by boiling in Lämmli buffer with and without β-Mercaptoethanol and analyzed by SDS-PAGE followed by Coomassie staining or Western blotting using an anti-S tag primary antibody (#71549 Sigma Aldrich).

**Determination of the nucleotide content**. For determination of the nucleotide content, 120 µl of 15 µM septin complex in solution were incubated for 5 min at 95 °C to denature the proteins. The precipitate was pelleted by centrifugation for 10 min at 16,100 xg. 110 µl of the supernatant was adjusted with 20 mM Tris-HCl pH 8.0 to a final volume of 5.5 ml and the nucleotide content in the solution was determined by analytical anion exchange chromatography on a MonoQ HR 5/5 column (Amersham Pharmacia), previously calibrated with known concentrations of GDP, GTP and GTPγS.

Nucleotide was eluted from the column with a linear NaCl gradient (0-450 mM NaCl) over 18 column volumes. The amount of eluted guanine nucleotides was quantified by calculating the integral of the corresponding peaks of the chromatogram in Origin v. 2021b (OriginLab). Determination of the relative nucleotide content per septin subunit was performed by using

the following formula correcting the protein concentration for the absorbance of the associated guanine nucleotides at 280 nm:

$$\text{nucleotide content} = \frac{[GDP] + [GTP]}{4 * (\varepsilon_s * 15 * 10^{-6} M - \varepsilon_g * ([GDP] + [GTP]))} * \varepsilon_s$$

with [GDP] and [GTP] representing the respective determined nucleotide concentration, $\varepsilon_s$ and $\varepsilon_g$ standing for the extinction coefficient at 280 nm of the stochiometric tetrameric septin complex (113570 M$^{-1}$*cm$^{-1}$) and the guanine nucleotide species (7720 M$^{-1}$*cm$^{-1}$)[39], respectively.

**Crystallization, structure determination- and analysis**. Septin complexes were crystallized by sitting-drop vapor diffusion. Well-diffracting crystals were obtained with 1.35 µL of protein solution (2 mg/mL) mixed with 1.35 µL of a reservoir condition containing 20% PEG 5000, 300 mM ammoniumsulfate and 100 mM Bis-Tris pH 6.5 and 0.3 µL of a seeding solution. The crystallization plates were incubated at 20 °C and crystals appeared within one day. Prior to flash-freezing in liquid nitrogen, crystals were cryoprotected in 10% 2,3-butandiol. The diffraction experiments were carried out at the ID30B beamline of the ESRF (https://doi.org/10.15151/ESRF-ES-928402160) at 100 K at a wavelength of 0.88560 Å. Processed X-ray diffraction data were obtained from the ESRF autoprocessing pipeline using XDS[40] as part of autoPROC Staraniso[41].

The structure was solved by molecular replacement in Phaser[42] using AlphaFold[25] models for Cdc3, Cdc10, Cdc12 and Shs1. The final model was built by iterative rounds of automated refinement and model building using Phenix[43] and Coot[44].

In automated refinement cycles xyz-reciprocal and -real space refinement as well as grouped B-factor refinement was performed. TLS refinement was included in the final stage of the refinement process using entire chains as TLS groups. Given the low-resolution regime of the collected diffraction data, further geometric restraints were defined[45] for automated refinement including secondary structure restraints, Ramachandran restraints and reference coordinate restraints to the initial structure of a refinement round. The number of refinement cycles per round was rigorously adjusted to yield the optimum R$_{free}$ improvement under maintenance of a small R$_{work}$-R$_{free}$ gap. Overall, this procedure resulted in improved geometric statistics, a lower clash-score and reduced values for R$_{free}$ and the R$_{work}$-R$_{free}$ gap. The final model showed 97.31% Ramachandran favored residues, 2.69% in allowed regions and 0% Ramachandran outliers, and a clash-score of 6.80 as evaluated from MolProbity in Phenix.

High-quality images of molecular structures were created with PyMOL (Schrödinger) and PyMOL was also used to calculate the relative solvent accessible surface area (SASA) of single residues in the hydrophobic crest. Mutations in the Cdc10 interface were assessed using FoldX 5.0[46].

**Reconstitution of septin complexes using AlphaFold-Multimer**. AlphaFold 2.1.2 was operated as paralleled version (ParaFold)[47] on the high-performance computing cluster bwFor JUSTUS2 at Ulm University in a custom installation allowing to adjust all input parameters. The AlphaFold-Multimer algorithm[22] was employed for the reconstitution of the tetramer and the different dimers. Cdc10(VL), Cdc3(Q81-A410), Cdc12(M1-E320), Cdc11(M1-S297) and Shs1(M1-E345) were provided as input sequences in the respective combinations in one single fasta file. Computing jobs on JUSTUS2 were scheduled using Slurm scripts and the analysis output files of the AlphaFold prediction algorithm were extracted using AlphaPickle (Arnold, M. J. (2021) https://doi.org/10.5281/zenodo.5708709). The generated pLDDT

and pAE scores were assessed to judge the quality of the prediction. The generated structure models were ranked according to their weighted pTM/ipTM scores[22] and only the model with the highest score was considered for further analysis.

**In silico design of artificial disulfide bridges**. Disulfide bond engineering sites were selected by introducing the AlphaFold model of the Cdc10-Cdc10 NC-dimer into the SSbondPre web server (http://liulab.csrc.ac.cn/ssbondpre)[27]. We selected three different inter-chain crosslinking positions at different contact sites for further analysis. All engineering sites were predicted with a high probability score resulting in the combined cysteine mutations Q21C and Q281C (99.5%), A132C and L300C (99.3%) as well as T19C and E23C (87.2%).

**Statistics and reproducibility**. For the nucleotide content assay two independent protein preparations were performed and each was assayed in triplicate. All analytical size exclusions were performed in triplicate, each replicate from an individual protein preparation. The droplet assay was performed in triplicate (three individual assays).

**Reporting summary**. Further information on research design is available in the Nature Portfolio Reporting Summary linked to this article.

## Data availability

The X-ray crystallographic data are available in the PDB under the identifier 8PFH. Numerical source data underlying the figures and mass spectrometric raw data are available for download on Zenodo (https://doi.org/10.5281/zenodo.10245882). Unprocessed, uncropped Western blots are shown in Supplementary Fig. 7. Generated plasmids and yeast strains are available from the corresponding author upon request.

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

## Acknowledgements
The authors thank Julian Baur, Reinhild Rösler and Stefanie Timmermann for technical assistance and the staff at beamline ID30B at the European Synchrotron ESRF for excellent assistance with data collection and automated processing. Support by the state of Baden-Württemberg through bwHPC and the German Research Foundation (DFG) through grant INST 40/575-1 FUGG (JUSTUS 2 high performance computing cluster) is acknowledged. Lukas Denkhaus was supported by DFG grant CRC 1384 (project ID 403222702).

## Author contributions
B.G. performed experiments, built the structure, analyzed the data and contributed to the manuscript. L.D. crystallized the protein, collected and processed the dataset and made the molecular replacement. S.G. analyzed the data. M.V. performed experiments. N.J. analyzed the data, designed research and improved the manuscript. T.G. performed experiments, analyzed the data, designed research and wrote the manuscript.

## Funding

## Competing interests
The authors declare no competing interests.
