## [Peer Review File · Communications Biology]

Reviewers' comments:

Reviewer #1 (Remarks to the Author):

The assembly of septin complexes requires conserved interactions between adjacent guanine nucleotide-binding domains (G-interface) and the N- and C-terminal regions (NC-interface) of septin monomers. The authors present a valuable crystal structure of a tetrameric septin complex (Shs1-Cdc12-Cdc3-Cdc10) from yeast. They show that G-interfaces replicate the conserved interactions known from human septins, all in the presence of GDP. By combining information from septin assembly predictions, crystal structures (from this work and other's) and experimental validation, they provide convincing data to show that an array of conserved hydrophobic residues, termed hydrophobic crest (ranging from the hook loop over the α 0-helix to the conserved Phe(β 1)) is indispensable for the NC-interface integrity. This work may be of interest to structural biologists and biochemists who study septins and beyond. I have some comments and suggestions to improve the manuscript, as follows:

1. Protein expression/purification

1.1. Please indicate the protein bands identity in the SDS PAGE gel (supplementary figure).

2. Tetrameric septin complex - crystallographic data quality and model refinement

2.1. The authors claim that the overall resolution of the crystal structure is 3.2 Å with anisotropic data showing an ellipsoidal completeness of 88.0%, but the highest resolution shell has a completeness of 36.4%, I/sigma of 1.5 and CC1/2 of 0.579. How much do the high resolution reflections contribute to the model refinement? It might be interesting to show some representative electron density figures (and B-factors) of the protein-protein interfaces of interest (Figures 3, 5 and 6) of the final model in the supplemental material. This would give the reader a better sense of how reliable these regions of the model are.

2.2. Also, the authors used automated cycles of refinement including TLS refinement and further geometric restraints to improve geometric statistics, the clash-score and to reduce values of Rfree and the Rwork-Rfree gap. Please discuss how overfitting was avoided given that Rfree/Rwork = 0.2897/0.2822 in Table 1.

2.3. Table 1: "multiplicity" is repeated.

3. Tetrameric septin complex – G interfaces

3.1. Figure 1 – Please indicate in the figure the relevant "structural features of small GTPases like the P-loop and the distinct switch 1 and switch 2 loops including the invariant DXXG-motif".

3.2. In the sentence: "Four residues from the P-loop are in contact with the beta-phosphate of the GDP (Fig. 4A)" – Fig. 4A should be Fig. 3A.

3.3. The following sentences are missing either a reference or a multiple sequence alignment to show the conservation of the sequence motifs:

"The P-loop consensus sequence in all four subunits is G1-L/T/I2-G3-K4-T/S5-T/A6."

"Shs1 is unique among the yeast septins in having a methionine in the first position of the otherwise strictly conserved DXXG motif in the switch 2 loop."

3.4. "We propose that Shs1 and Cdc3 have either a higher affinity for GDP impeding GTP uptake or that they hydrolyze GTP via another mechanism than the loaded spring mechanism".

Is there any evidence for "a higher affinity for GDP" than GTP for yeast Shs1 and Cdc3 (or their human homologs)?

4. AlphaFold models

4.1. It would be helpful to make the complete AlphaFold models available as supplementary material. Also, intra- and inter-domain prediction accuracy should be further discussed beyond RMSD-Ca estimation. What about amino acid side-chain modelling?

4.2. Predicted coiled-coil interactions are not discussed (or validated) for the Cdc10-Cdc10 interface. Can they interfere with the other protein-protein contact surfaces?

4.3. Octamer curvature – Please estimate the bending angle (figure 4), discuss its consequences and compare with previous literature.

5. NC interface

5.1. “Since the hydrophobic effect is weaker in the terminal NC interface, the contribution of a salt bridge network might become more important for its stability”.

It would be informative to evaluate the salt concentration effect on yeast septin NC interfaces in vitro.

Reviewer #3 (Remarks to the Author):

Grupp et al 2023 Communications Biology

Grupp et al describe the structure of the yeast septin octamer based on a tetrameric Shs1-Cdc12-Cdc3-Cdc10 complex solved by x-ray crystallography and AlphaFold-Multimer modeling. Their structure allowed identification of a new structural feature in the alpha0-helix, the “hydrophobic crest,” which they show is important for NC interface interactions in human septins too.

A major strength of this paper is the elegant use of disulfide bond formation to test the Cdc10-Cdc10 NC interface model generated by AlphaFold-Multimer. The authors also test their octamer model through reintroduction of the alpha-0-helix to truncated versions and show restoration of assembly. The work also includes thorough experimental verification of GDP/GTP levels and investigation of multiple mutants throughout.

As the authors point out, the idea of an alpha0-helix hydrophobic crest inserting into the hydrophobic core of neighboring subunits to stabilize the NC interface is consistent with known information of septin assembly from multiple species and could resolve some long standing questions about assembly of heteropolymers.

My biggest criticism of the manuscript is that it was somewhat difficult to read. It presents a lot of detailed structural information in the context of known septin domains and motifs, but does not orient the reader to the domains. It would be very helpful to the reader to have a figure that pulls together structural features and known conserved domains. I referred frequently to Cavini et al 2021 Fig 3 to follow the descriptions in this manuscript. Perhaps a similar kind of overview figure could be added?

A few small errors spotted:

- 1) Pg3, paragraph 3. “Reconstitution experiments show that Shs1-rods from curved fibers below membranes...” Did the authors mean “form”?
- 2) P24, Fig 2 legend. “The inset shows a Coomassie stained SDS-PAGE gel” Did the authors mean “inset”?
- 3) Pg5, p 3: “The P-loop consensus sequence in all four subunits is G1-L/T/I2-G3-K4-T/S5-T/A6. Four residues from the P-loop are in contact with the b-phosphate of the GDP (Fig. 4A).” Did the authors mean Fig. 3A?
- 4) Pg7, top paragraph: “A PDB file of the predicted Cdc10 dimer is provided as supplementary structure.” Supplementary structure or link not provided.

Reviewer #4 (Remarks to the Author):

The authors report the x-ray structure of GDP-bound heterotetrameric septin complex comprising three canonical subunits (Cdc10, Cdc3, Cdc12), and a non-canonical subunit, Shs1, thought to impart curvature to higher-order septin assemblies. The asymmetric unit comprises a single heterotetramer that lacks the Cdc10 "hydrophobic crest" required for assembly of octamers and other higher-order structures; inclusion of that element would have almost certainly hindered crystallization. The authors suggest that crystallographic contacts among heterotetramers are close to those in an octameric assembly. They show rigorously using mutagenesis and crosslinking between engineered cysteines that this crystallographic contact occurs in solution. These data are further corroborated using AlphaFold 2. They then show using size exclusion chromatography that inclusion of the Cdc10 "hydrophobic crest" facilitates formation of octamers by increasing the affinity between heterotetramers via the crystallographic contact. They show that the features of the hydrophobic crest are conserved in human septins and show using size exclusion chromatography that the hydrophobic crest drives octamer formation of human septins just as it does in *Saccharomyces cerevisiae*.

The Authors report first high-resolution account of a higher-order septin assembly, and show convincingly that the assembly mechanisms are conserved from yeast to humans. I enjoyed reading the paper and it's my view that this work rigorously supports the authors claims and represents a significant contribution to the field.

I do have just a few minor suggestions, none of which involve additional experiments:

1. I was a little puzzled by the exceedingly small R/R-free gap. While appreciate that this should be a GOOD thing, as it suggests the absence of overfitting/over-refinement, it's unusual for sub-3Å x-ray structures. I refined the structure myself using phenix using individual B-factors, TLS (1 group/chain), weight optimization, and secondary structure restraints and saw a dramatic drop in R while R-free remained at ~0.29. This isn't to suggest that the x-ray structure doesn't support the author's conclusions, it's more of a curiosity that I think warrants a bit of discussion.
2. In the abstract and elsewhere in the manuscript the authors state that "AlphaFold-multimer allows us to eventually reconstruct the complete.....". That makes it sound like it's not finished yet, but will be, "eventually"... Simply say something like "We used AlphaFold to model an octameric Septin Rod". You can also remove "in-silico"; It's redundant after having said AlphaFold.
3. The paper is exceedingly clear, which I appreciated; This is the first time I've reviewed a manuscript, and I have a feeling that it's not always this easy/enjoyable. Despite the overall clarity and conciseness, there are bits of language here and there that I think could be rephrased to make it sound a bit more elegant. This is entirely optional, and perhaps something that the editors will help with.

Great work. I learned a lot about septins and was particularly impressed with the rigorous validation of a crystallographic interface.

Manuscript COMMSBIO-23-2476-T
Point-by-point answer to the reviewers

General remarks:

At the time of submission of the initial manuscript, a crystal structure comprising the central tetramer of the septin rod Cdc3-10-10-3 was published. We acknowledge this structure in the revised manuscript (lines 247-252 and Suppl. Fig.5).

We removed the AF reconstructed Cdc11 containing octamer from Fig.4. as we prefer to focus on the Shs1 complex in this figure.

Reviewer 1

1.1. Please indicate the protein bands identity in the SDS PAGE gel (supplementary figure).

In Suppl. Fig. 1A the septin protein bands are labelled with identifiers.

2.1. The authors claim that the overall resolution of the crystal structure is 3.2 Å with anisotropic data showing an ellipsoidal completeness of 88.0%, but the highest resolution shell has a completeness of 36.4%, I/sigma of 1.5 and CC1/2 of 0.579. How much do the high resolution reflections contribute to the model refinement? It might be interesting to show some representative electron density figures (and B-factors) of the protein-protein interfaces of interest (Figures 3, 5 and 6) of the final model in the supplemental material.

Weak reflexes of the highest resolution shell contribute to the resolution features of the structure but we did not quantify this effect. However, we included the respective references in the caption to Table 1.

Representative electron density figures are now presented in Suppl. Fig. 2, which is also announced in the text (line 138).

2.2. Also, the authors used automated cycles of refinement including TLS refinement and further geometric restraints to improve geometric statistics, the clash-score and to reduce values of Rfree and the Rwork-Rfree gap. Please discuss how overfitting was avoided given that Rfree/Rwork = 0.2897/0.2822 in Table 1.

We used reference coordinate restraints to maintain a small $R_{work}-R_{free}$ gap which avoids overfitting. This was already stated in the Methods section. We extended this part to make this clearer to the reader (lines 554-561).

2.3. Table 1: "multiplicity" is repeated.

Corrected.

3.1. Figure 1 – Please indicate in the figure the relevant "structural features of small GTPases like the P-loop and the distinct switch 1 and switch 2 loops including the invariant DXXG-motif".

We included another panel in Fig. 1 which highlights relevant structural features of septins.

3.2. In the sentence: “Four residues from the P-loop are in contact with the beta-phosphate of the GDP (Fig. 4A)” – Fig. 4A should be Fig. 3A.

Corrected.

3.3. The following sentences are missing either a reference or a multiple sequence alignment to show the conservation of the sequence motifs:

“The P-loop consensus sequence in all four subunits is G1-L/T/I2-G3-K4-T/S5-T/A6.”

“Shs1 is unique among the yeast septins in having a methionine in the first position of the otherwise strictly conserved DXXG motif in the switch 2 loop.”

We did not state that the P-loop consensus sequence is conserved. This is rather the consensus sequence of the four subunits in our structure. We pointed this out in the revised text (line 163).

Regarding the DXXG motif in Shs1 we inserted a suitable reference (line 174).

We furthermore provide consensus sequences of selected important structural motifs in another panel to Fig. 1.

3.4. “We propose that Shs1 and Cdc3 have either a higher affinity for GDP impeding GTP uptake or that they hydrolyze GTP via another mechanism than the loaded spring mechanism”. Is there any evidence for “a higher affinity for GDP” than GTP for yeast Shs1 and Cdc3 (or their human homologs)?

Experimental data are available for the human SEPT2 only. We added the respective information incl. reference to the discussion (lines 414-416).

4.1. It would be helpful to make the complete AlphaFold models available as supplementary material. Also, intra- and inter-domain prediction accuracy should be further discussed beyond RMSD-Ca estimation. What about amino acid side-chain modelling?

All AF models are now provided as supplementary data set. We performed an additional “all-atom RMSD” estimation. These corresponding values are now included in Suppl. Table 1.

4.2. Predicted coiled-coil interactions are not discussed (or validated) for the Cdc10-Cdc10 interface. Can they interfere with the other protein-protein contact surfaces?

The central subunits of the octamer(s) (SEPT9 in human and Cdc10 in yeast) do not possess coiled coils. We amended the introduction in this respect to make this point clear to the reader (lines 88/89).

4.3. Octamer curvature – Please estimate the bending angle (figure 4), discuss its consequences and compare with previous literature.

We complemented the manuscript with a brief discussion of the recently published Cdc3-10-10-3 structure. We assembled also an octamer using the experimental Cdc10-Cdc10 interface (Suppl. Fig 5). This octamer has a different bending angle than the one with the

predicted central Cdc10 interface. The bending of the octamer is a nice feature, concomitant with the literature, but putting any quantitative number (by means of a bending angle) and any discussion of such quantitative features would result in pure speculation. The function of octamer bending would point towards the ability of septins to shape biological membranes, a topic that we do not cover in the framework of this manuscript. Including this field here would require an additional detailed discussion which in our opinion is beyond the scope of this manuscript.

Taken together, we refrain from including and discussing a bending angle.

5.1. "Since the hydrophobic effect is weaker in the terminal NC interface, the contribution of a salt bridge network might become more important for its stability". It would be informative to evaluate the salt concentration effect on yeast septin NC interfaces in vitro.

The salt concentration effect on filament formation was already evaluated by us and others in *in vitro* experiments (Renz et al, 2011, Bertin et al. 2009). These results are mentioned and cited in the discussion (line 465). The respective paragraph was reworded in the framework of the revision.

It would be additionally interesting to evaluate this effect specifically for the terminal NC-interface by forming an "inside out" octamer Cdc10-3-12-11-11-12-3-10. These experiments are currently ongoing in our laboratory in the context of another project. We are afraid we cannot include them in this manuscript.

Reviewer 3

My biggest criticism of the manuscript is that it was somewhat difficult to read. (...) It would be very helpful to the reader to have a figure that pulls together structural features and known conserved domains.

We included a respective panel in Fig. 1. See comment to point 3.1 of reviewer 1.

1) Pg3, paragraph 3. "Reconstitution experiments show that Shs1-rods from curved fibers below membranes..." Did the authors mean "form"?

2) P24, Fig 2 legend. "The inlet shows a Coomassie stained SDS-PAGE gel" Did the authors mean "inset"?

3) Pg5, p 3: "The P-loop consensus sequence in all four subunits is G1-L/T/I2-G3-K4-T/S5-T/A6. Four residues from the P-loop are in contact with the b-phosphate of the GDP (Fig. 4A)." Did the authors mean Fig. 3A?

All minor issues 1-3 raised by this reviewer were addressed.

4) Pg7, top paragraph: "A PDB file of the predicted Cdc10 dimer is provided as supplementary structure." Supplementary structure or link not provided

We hope that all uploaded files are fully accessible to the reviewers.

Reviewer 4

I was a little puzzled by the exceedingly small R/R-free gap (...)

See comments to point 2.2 by Reviewer 1.

In the abstract and elsewhere in the manuscript the authors state that "AlphaFold-multimer allows us to eventually reconstruct the complete.....". That makes it sound like it's not finished yet, but will be, "eventually"... Simply say something like "We used AlphaFold to model an octameric Septin Rod". You can also remove "in-silico"; It's redundant after having said AlphaFold.

The abstract and the introduction were reworded following the reviewer's suggestion (line 48 and 118/119).

REVIEWERS' COMMENTS:

Reviewer #1 (Remarks to the Author):

Thank you for the explanations. The authors have submitted an improved manuscript.

Reviewer #3 (Remarks to the Author):

The Authors have satisfactorily addressed My comments.